# Fractional Excretion of Phosphate (FeP) Is Associated with End-Stage Renal Disease Patients with CKD 3b and 5

**DOI:** 10.3390/jcm8071026

**Published:** 2019-07-12

**Authors:** Antonio Bellasi, Lucia Di Micco, Domenico Russo, Emanuele De Simone, Mattia Di Iorio, Raffaella Vigilante, Luca Di Lullo, Biagio Raffaele Di Iorio

**Affiliations:** 1Department of Research, Innovation and Brand Reputation, ASST Papa Giovanni XXIII, 24127 Bergamo, Italy; 2Nefrology and Dialysis, Ospedale del Mare, ASL Napoli 1 Centro, 80147 Napoli, Italy; 3Department of Nephrology, School of Medicine, University “FEDERICO II”, 80131 Napoli, Italy; 4Nefrology and Dialysis, AORN “San Giuseppe Moscati”, 83100 Avellino, Italy; 5Data Scientist, Cardarelli’s Nephrology Consultant, 83100 Avellino, Italy; 6Department of Translational Medical Sciences, University of Campania “Luigi Vanvitelli”, 80121 Naples, Italy; 7Department of Nephrology and Dialysis, Ospedale Parodi, Delfino, 00034 Colleferro (Rome), Italy; 8Nephrology and Dialysis, AORN “Antonio Cardarelli”, 80131 Napoli, Italy

**Keywords:** phosphate, phosphate balance, CKD–MBD, outcome, fractional excretion of phosphate, FeP

## Abstract

**Background**: The perturbation of phosphate homeostasis portends unfavorable outcomes in chronic kidney disease (CKD). However, the absence of randomized clinical trials (RCT) fuels the discussion of whether phosphate or some other phosphorous-related factor(s) such as fibroblast growth factor 23 (FGF-23) mediates the cardiovascular and systemic toxicity. We herein test whether the fractional excretion of phosphate (FeP) as a marker of renal stress to excrete phosphorous predicts unfavorable outcomes in CKD patients. Methods: Retrospective, cross-sectional observational study. For current analysis, an historical cohort of 407 records of CKD stage 3b-5 patients attending between January 2010 and October 2015 at the Nephrology Unit of Solofra (AV), Italy were utilized. Demographic, clinical, laboratory, and outcome data were identified through the subjects’ medical records. We tested whether quartiles of FeP are associated with the risk of CKD progression or all causes of death. Parametric as well as non-parametric tests, linear and logistic regression, as well as survival analysis were utilized. Results: Overall, we investigated middle-age (mean 66.0, standard deviation 12.3 years) men and women (male 43%) with CKD stage 3b to 5 (creatinine clearance 32.0 (13.3) mL/min). Older age, lower diastolic blood pressure, poor renal function, as well as higher serum phosphate were associated with FeP. Patients with higher FeP were at an increased risk of starting dialysis or dying (hazard ratio 2.40; 95% confidence interval (1.44, 3.99)). Notably, when the two endpoints were analyzed separately, FeP was associated with renal but not all-cause survival. **Conclusion**: FeP is associated with ESRD, but not all-cause mortality risk in a large cohort of moderate to advanced CKD patients. Future efforts are required to validate FeP as a marker of nephron stress and risk factor for CKD progression in this high-risk population.

## 1. Introduction

The perturbation of phosphate homeostasis portends unfavorable outcomes in chronic kidney disease (CKD) [1]. Indeed, a large body of evidence supports the notion that higher levels of serum phosphate are associated with risks of all-cause and cardiovascular (CV) mortality or CKD progression [1,2,3]. However, the absence of randomized clinical trials (RCT) fuels the discussion on whether phosphate or some other phosphorous-related factor(s) such as fibroblast growth factor 23 (FGF-23) mediates the cardiovascular and systemic toxicity [4,5]. 

The biological mechanisms that link phosphorous with unfavorable outcomes are still partially unclear [6,7]. In vitro evidence demonstrates that vascular smooth muscle cells exposed to progressively higher concentrations of phosphate differentiate, acquire an osteoblastic-like phenotype, and prompt vascular calcification deposition and progression by secreting a bone matrix within the context of the arterial wall [8,9]. This hypothesis is corroborated by the observation of a strong association of serum phosphate and cardiovascular calcification, as well as arterial stiffness in different stages of CKD [10,11,12]. Other studies suggest that hyperphosphatemia is linked to a suboptimal response to antiproteinuric treatments such as angiotensin-converting enzyme (ACE) inhibitors and low-protein diet [3,13,14]. 

If data suggest that hyperphosphatemia is undesirable, the optimal range for serum phosphate is still unknown [6], and some lines of evidence suggest that demographic as well as clinical characteristics may modulate the risk associated with hyperphosphatemia [1,4,15]. Indeed, epidemiological studies document different thresholds above which serum phosphate is associated with a significant increase risk of fatality [6]. A plausible explanation to reconcile what has been shown by observational studies is that balance rather than serum or urinary levels of phosphate is what modulates the risk associated with phosphate metabolism abnormalities [6]. Since serum levels of phosphate poorly reflect phosphate balance in normal as well as impaired renal function [16,17], a considerable effort is devoted to defining new markers of phosphate homeostasis and testing their associations with outcome.

Fractional excretion of phosphate (FeP) is a measure of the fraction of the amount of phosphate filtered by the glomerulus, which is excreted into urine. FeP is expressed as the ratio of the clearance of phosphate to creatinine clearance, and it has been postulated that excreting phosphate is a marker of nephron stress [17,18]. In subjects with preserved renal function, FeP levels below 20% are considered normal. However, several factors such as excessive dietary load and degree of renal dysfunction may affect FeP, and the clinical relevance of FeP in CKD is far from being established [17,18]. We sought to investigate the association of FeP as a marker of phosphate balance with relevant outcomes in a large cohort of patients with moderate to advanced CKD.

## 2. Experimental Section

For this retrospective study, an historical cohort of 407 records of CKD stage 3b to 5 patients attending between January 2010 and October 2015 at the Nephrology Unit of Solofra (AV), Italy were utilized. All the consecutive adult patients referred to the unit were included in this retrospective analysis. The study reflects standard practice in CKD care, and it was notified to the local institutional Ethical Review Board on 12 December 2009.

### 2.1. Data Collection

Demographic and clinical data were identified through the subject medical record. In current analyses, demographic, clinical, and laboratory data at the time of nephrology referral were collected. Similarly, outcome data were also collected. All the pieces of information were collected in the study database and de-identified. A history of cardiovascular disease (CVD) was defined as a history of myocardial infarction, stroke, coronary artery bypass surgery, angioplasty, and peripheral arterial disease. Diabetes mellitus was defined as a self-reported medical condition or by a fasting glucose equal to or greater than 126 mg/dl or the use of hypoglycemic drugs. Hypertension was defined as a self-reported medical condition or the use of antihypertensive drugs. Renal function was assessed as 24-h creatinine clearance (CrCl), while the glomerular filtration rate (GFR) was estimated by the standard four-variable MDRD (Modification of Diet in Renal Disease) formula [19].

Fasting routine blood laboratory and 24-hour urine collection were used. Samples were collected at the time patients were referred to the nephrology clinic as part of the standard care. Creatinine in plasma and urine was measured by means of modified kinetic Jaffè reaction serum levels of albumin as well as serum and urinary levels of urea, sodium, phosphate, and potassium, which were measured by an autoanalyzer (Abbott Architect, Italy). The parathyroid hormone (PTH) level was assessed by two radiolabeled purified antibodies specific for two different regions of the PTH molecule (PTH 1–34 and PTH 39–84 (Beckman Coulter, Unicel DXL 800, Italy). Finally, hemoglobin was measured by Coulter counter (Sismex, Dasit, Italia).

Since the fractional excretion of phosphate (FeP) based on spot urine samples are not likely to reflect the daily phosphate balance [17], we calculated FeP using 24-h urine phosphate and creatinine samples, which are less sensitive to the circadian variations of urinary excretion. The fractional excretion of phosphate (FeP) was calculated using the following formula: FeP = (24-h urine phosphate × serum creatinine) × 100/(serum phosphate X 24-h urine creatinine). Protein intake was calculated according to the following formula: Protein intake = 6.25* (0.46* UUN + 0.031* ideal body weight) + PPU (where UUN is urinary urea and PPU is proteinuria).

### 2.2. Outcome Assessment

For this study, we examined the association of fractional excretion of phosphate (FeP) and three outcomes: (i) the occurrence of the composite event of all-cause mortality and CKD progression to end-stage renal disease (ESRD) (defined as the need for renal replacement therapy), whichever came first; (ii) the occurrence of CKD progression to ESRD; and (iii) the occurrence of all-cause mortality. These are regarded as the most clinically relevant outcomes for CKD patients. However, in light of the association between fast renal function deterioration and risk of death [20], it is important to test the effect of FeP on the combination of the two outcomes before testing the effect of FeP on each of the two outcomes separately. Data on clinical outcomes were identified through the review of the study subject medical record.

### 2.3. Statistical Analysis

As the fractional excretion of phosphate (FeP) is not linearly associated with the clinical outcomes tested, the study cohort was grouped into four strands based upon quartiles of FeP at baseline. Hence, baseline demographics, medical history, and laboratory data were expressed and compared according to the quartiles of FeP. Continuous variables were expressed as a mean ± standard deviation (SD), and categorical variables were expressed as a number and percentage. Non-normally distributed variables are expressed as a median and interquartile range (IQR). Comparisons between groups were conducted by using the analysis of variance (ANOVA) for continuous variables and the Chi-square test for discrete variables. Since a FeP <20% is suggested as a normal value of FeP, linear and logistic regression were used to investigate the predictors of increased FeP (for logistic regression analysis, predictors of normal (FeP <20%) versus abnormal FeP (>20%) were tested). To gauge the association between FeP and the outcomes of interest, we first plotted the incidence of the endpoint curves according to FeP quartiles using the Kaplan–Meier product limit method, and we used the two-sided log rank test to calculate the overall *p* value for differences in (i) the risk of the composite outcome, (ii) renal survival, and (iii) all-cause survival. Next, we calculated the multivariable adjusted hazard ratios (HR) by fitting the data to Cox proportional hazard models. All of the models presented are unadjusted (Model 1), case mix-adjusted (Model 2: adjusted for age, history of CVD, history of diabetes mellitus, history of hypertension, systolic and diastolic blood pressure, and renal function), and lastly, fully adjusted (Model 3: model 2, protein intake, azotemia, albumin, serum phosphate, PTH, triglycerides, 24-h urinary potassium). All the variables that were forced into the survival models were selected a priori based on evidence suggesting an association with either FeP or the outcomes of interest or, alternatively, because of evidence of an association with FeP at univariable analysis (*p* < 0.1) (Table 1).

To investigate the independent and incremental ability of FeP to predict the risk of CKD progression, logistic regression analyses were performed to determine independent associations of high-FeP (i.e., defined as FeP >75th percentile of the study cohort distribution) and other variables. The stepwise method was used in the logistic regression analyses to identify the most parsimonious model to predict renal survival. Age, history of CVD, history of diabetes mellitus, history of hypertension, systolic and diastolic blood pressure, renal function, protein intake, azotemia, albumin, serum phosphate, PTH, triglycerides, and 24-h urinary potassium were entered the model as independent variables, and the outcome of interest was entered as a dependent variable. The incremental ability of the most parsimonious logistic model with or without high FeP to predict the risk of CKD progression was assessed with the C-index, calibration statistic, and integrated discrimination improvement (IDI) and continuous net reclassification improvement (NRI). The statistical significance level was set as a *p*-value < 0.05. All statistical analyses were conducted using R version 3.1.3 (2015-03-09) for Mac.

## 3. Results

Overall, we investigated middle-age (mean 66.0 (standard deviation 12.3) years) men and women (male 43%) with CKD stage 3b to 5 (creatinine clearance 32.0 (13.3) mL/min) (Table 1).

FeP distribution was highly skewed (mean (SD): 44 (30)%; median (interquartile range): 36 (24–55)%) and only a minority of study subjects (16%) presented a FeP below 20% (considered normal phosphate excretion) (Figure 1). 

At linear regression analysis, higher FeP was associated with older age, higher azotemia as well as lower diastolic blood pressure, creatinine clearance and serum phosphate (Table 2). Similar results were yielded by logistic regression analysis when predictors of abnormal FeP (defined as FeP > 20%) were tested. After adjustment for confounders, abnormal FeP (>20%) was inversely associated with creatinine clearance (B-0.03, *p* = 0.006), diastolic blood pressure (B-0.04, *p* = 0.01) and serum phosphate (B-0.43, *p* = 0.008) (Table 3).

Over mean follow-up of 45 months (min: 3, max 60), 134 and 54 persons died of all-cause or started dialysis.

At univariate analysis, a significant a graded increase in the risk of the composite outcome of all-cause mortality or progression to ESRD was noted (Figure 2A; Table 4A). Progressive adjustments for factors either associated with FeP or the outcomes of interest only minimally attenuated the strength of the association (Table 4A). FeP greater than 55% was associated with a 2.4 fold increase in the risk of either dying of all-cause or start dialysis (Hazard ratio (HR): 2.39; 95% Confidence Interval (95%CI): 1.44–3.99; *p* < 0.001). 

Similar findings were also noted when the risk of CKD progression to ESRD was modelled (Figure 2B; Table 4B). A significant and stepwise increase in the risk according to FeP quartile was noted irrespective of progressive adjustments (Table 4B). FeP greater than 55% was associated with a 12.3 fold increase in the risk of dialysis inception (HR: 12.3; 95% CI: 3.64–41.73; *p* < 0.001).

In contrast with previous findings, no association between FeP and time to all-cause mortality was noted (Figure 2C; Table 4C). Indeed, adjustment for factors either associated with FeP or the outcomes of interest progressively attenuated the strength of the association that was no longer detected after multiple adjustments (Table 4C).

To evaluate the incremental ability of FeP to predict renal survival, logistic regression analysis was utilized. As reported in Table 5, FeP predicted the risk of renal progression beyond renal function assessed through 24-h creatinine clearance. Indeed, the Area Under the Curve (AUC, Figure 3) for the model without and with high-FeP (Table 6) (were 0.770 (95% CI: 0.695–0.846) and 0.802 (95% CI: 0.734–0.869; *p* for comparison 0.0009), respectively.

## 4. Discussion

Our results lend further support to the notion that phosphate burden is linked to an adverse effect on renal function and prognosis in moderate-to-advanced CKD independently of factors either associated with FeP or the outcomes of interest. Furthermore, the stronger association of FeP with CKD progression rather than all-cause mortality as well as the independency of these associations with residual renal function suggest that phosphate burden may mediate renal damage that, in turn, partially contribute to the explain the effect of phosphate burden on the risk of fatality.

An expanding body of evidence supports the notion that an increase in phosphate excretion per nephron induces renal tubular damage and accelerates nephron loss. In a large observational study of 2,340 patients with non-dialysis dependent CKD (ND-CKD) under stable care in 40 Italian nephrology clinics, it was reported that serum phosphorous, albeit in the range of normality, was associated with the 24-h proteinuria and risk of CKD progression [14]. Of interest, the risk of ESRD associated with serum phosphate was progressively attenuated by greater degree of 24-h proteinuria [14]. Although these observations may be cofounded by unmeasured factors, a plausible explanation is that both phosphate and proteinuria mediated the tubulo-interstitial fibrosis in the kidney [18,21,22]. Hence, the risk of CKD progression associated with phosphate burden is less evident when overt proteinuria is established. This hypothesis is supported by the observations that the nephroprotective effects of various antiproteinuric therapies decrease when phosphorus levels are high [3,13]. Indeed, the antiproteinuric effect of both ACE-I and low protein diet seems to vanish as serum phosphate levels increase [3,13]. However, in contrast with these observations, 3-mon administration of Sevelamer, a non-calcium containing phosphate binder, failed to reduce proteinuria in a recently published cross-over randomized controlled study (RCT) [23] of 53 middle age (mean age 55 + 17), normophosphatemic (mean serum phosphate: 3.8 + 0.6) stage 3 CKD patients (mean GFR 49 + 23 mL/min/1.73 m^2^) with optimal renin-agiotensin-aldosterone (RAAS) system blockage [23].

In line with these results, RCTs that have investigated the impact of phosphate lowering agents on cardiovascular endpoints yielded disappointing results. Chue and coworkers [24], showed no effect of sevelamer vs. placebo on either pulse wave velocity (PWV) or left ventricular hypertrophy (LVH) in cohort of 103 CKD stage 3 patients with normal serum levels of phosphate. In another RCT, Block and coworkers [25] showed a worrisome increase in coronary artery calcification (CAC) in normophosphatemic CKD stage 3–4 subjects allocated to various phosphate binder regimens when compared to placebo. Although it is possible that phosphate is not causally related to clinical outcome, these RCTs recruited CKD patients with normophosphatemia and no information on phosphate balance has been reported. Administration of phosphate binders in these subjects had minimal or no effects on serum levels but a larger effect on urinary levels of phosphorous [23,24,25], suggesting that renal excretion is modulated according to phosphate balance rather than serum levels [6] and that subjects with normophosphatemia maybe exposed to positive as well as negative phosphate balances [6]. Indeed, overt hyperphosphatemia develops only if the number of residual and functional nephrons cannot compensate for the daily phosphate intake [6]. Accordingly, epidemiological studies have repeatedly shown that hyperphosphatemia tends to develop only late in the course of CKD, when a significant number of nephrons is lost [26].

In line with these observations, our results document that in a cohort of normophosphatemic CKD stage 3b, 4 and 5 only about one quarter of the selected individuals exhibit normal FeP (Table 1). However, we expand previous reports by suggesting that the risk associated phosphate burden maybe predicted by FeP calculated using the 24 h urine collection. Similar conclusions were drawn by the Heart and Soul study investigators. In a cohort of 872 subjects with preserved renal function (eGFR 60–70 mL/min/1.73 m^2^) and normal serum levels of phosphorous (3.5–3.8 mg/dL), the risk of all-cause and CV mortality was predicted according to FGF23 and FeP levels [4]. Of note, a significant effect interaction between FeP and FGF23 was also present, suggesting that phosphaturia, triggered by high FG23, may represent a physiological response to a positive phosphate balance and increased FeP a sign that this adaptive mechanism has become maladaptive. Indeed, a significant tubular stress on the residual functional nephrons is deemed necessary to excrete phosphate and compensate for the nephrons loss in CKD.

Since phosphatemia does not reflect phosphate balance and increased risk of vascular calcification as well as mortality has been described even in normophosphatemic subjects [6], it is plausible that other factors such as the calciprotein particles (CPPs) contribute to the CV and all-cause mortality risk. CPPs are defined as aggregates of serum proteins (mainly fetuin-A, FGF23) and calcium-phosphate crystals that are indispensable for phosphate solubilization in serum as well as prevention of calcium-phosphate crystals precipitation in the extraosseous tissues [22]. However, CPPs promote inflammation, vascular endothelial death, arterial stiffness, vascular smooth muscle cell calcification as well as tubular damage [22]. Hence, CPPs can potentially mediate the risk associated with phosphate burden and explain the poor correlation of serum levels and phosphate balance [22].

Twenty-four hours rather than spot urine urine phosphaturia is a more accurate marker of the daily phosphate intake since it is less influenced by the circadian rhythm of phosphorous excretion [27]. With the exception of one study in renal transplanted patients [28], no association between 24-h urine phosphate and renal outcome has been documented in CKD patients [17] possibly due to the poor correlation with the stress needed to excrete the excessive phosphate load of the residual renal mass. Hence, for current analysis, we calculated the fractional excretion of phosphate (FeP) using the 24-h phosphaturia because is less sensitive than spot urine to the circadian variations of urinary excretion of phosphate and it may represent an indirect marker of the nephron stress to excrete unnecessary phosphate load. In these regards, Kawasaki and coworkers [29] showed a stepwise increase in the risk of CKD progression (defined as a composite of ESRD and 50% reduction of eGFR) according to quartiles of the ratio of 24-h urinary phosphorous excretion per creatinine clearance (24U-P/Cr) in a cohort of 191 CKD stage 4 patients (mean eGFR: 19 + 13 mL/min/1.73 m^2^) [29]. After adjustments for multiple factors, the highest quartiles of 24UP/Cr was associated with a 7.89 fold increased risk of CKD progression (HR 7.89; 95%CI: 1.74–44.33) [29]. In another study of 95 patients with CKD stage 2, 24-h phosphaturia corrected for creatinuria predicted eGFR decline independently of confounders [30]. Of interest, the animal model of uninephrectomized rats utilized to shed light on the potential mechanisms responsible for the clinical findings, showed that high phosphate diet increased phosphaturia promoting renal tubular damage, inflammation, oxidative stress and low klotho expression [30]. Finally, in another series of 273 kidney transplant recipiens abnormal FeP (defined as >20%) and elevated PTH (defined as 90 pg/mL) were the only two significant markers of bone mineral metabolism associated with graft function loss after adjustment for confounders [28].

Our study is not without limitations. This is an observational study and despite of our best effort to control for factors either associated with eGFR or mortality, some unmeasured residual confounding factors cannot be excluded. Indeed, nutritional patterns or use of phosphate binders before and after the referral to the Nephrology Unit were not investigated. However, the robustness of the estimated HR after adjusting for multiple co-variables reduce the probability of bias in our data. Furthermore, the cross-sectional and retrospective nature of the study prevents us clarifying the causal relationship between FeP and mortality or CKD progression in this cohort of individuals. However, the homogeneous study cohort and patient care increase the plausibility of current results and should prompt future studies to test the clinical utility of FeP as a marker of nephron stress and risk for renal outcome. Similarly, the cross-sectional nature of study does not allow to investigate the impact of changes of FeP on outcomes. Finally, the lack of data on newer markers of phosphate homeostasis (i.e., fibroblast growth factor 23) or data on echocardiogram parameters precludes any inference on the potential relationship of FeP and heart failure and future efforts are required to shed light on this aspect.

In conclusion, FeP is associated with ESRD but not all-cause mortality risk in a large cohort of moderate to advanced CKD patients. Future efforts are required to validate FeP as a marker of nephron stress and risk factor for CKD progression in this high-risk population.

## Figures and Tables

**Figure 1 jcm-08-01026-f001:**
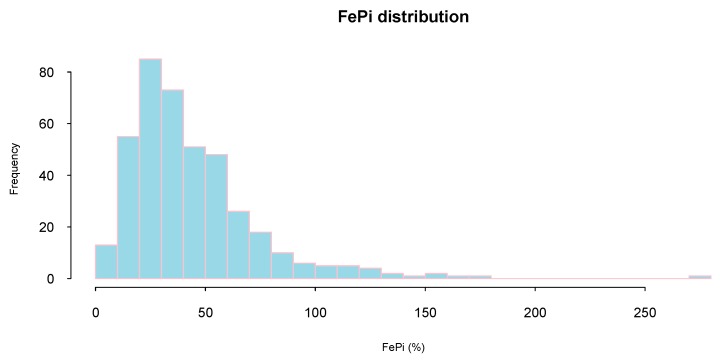
Fractional Excretion of Phosphate (FeP) distribution.

**Figure 2 jcm-08-01026-f002:**
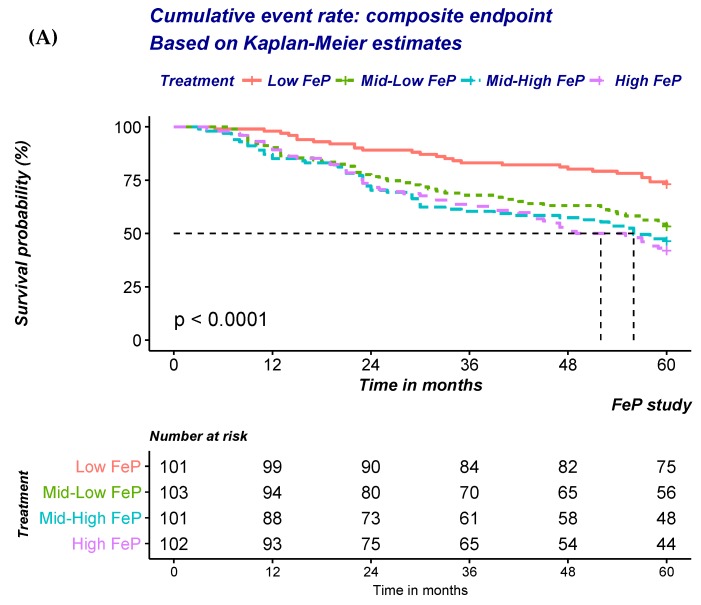
Overall likelihood of experiencing (**A**) the composite end point, (**B**) progression to ESRD, (**C**) all-cause mortality, according to quartile of Fractional Excretion of Phosphate (FeP) based on the Kaplan-Meier estimates.

**Figure 3 jcm-08-01026-f003:**
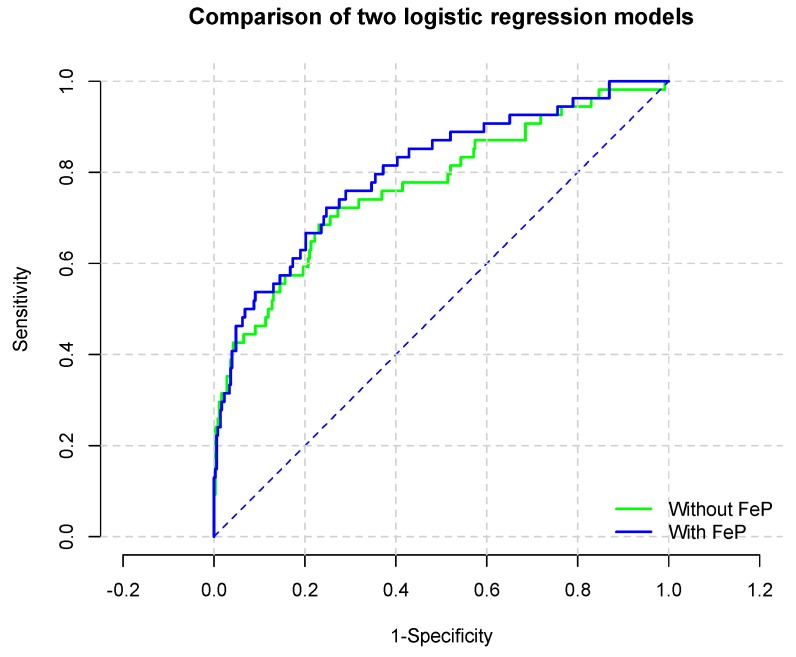
Receiving Operator Curve of the most parsimonious model with and without high- Fractional Excretion of Phosphate (FeP) to predict CKD progression.

**Table 1 jcm-08-01026-t001:** Demographic, clinical, and laboratory characteristics of the study cohort according to quartiles of the fractional excretion of phosphate (FeP). BMI: body mass index, CKD: chronic kidney disease, PTH: parathyroid hormone.

	Total (*n* = 407)	Low Quartile (*n* = 102)	Mid–Low Quartile (*n* = 102)	Mid–High Quartile (*n* = 101)	High Quartile (*n* = 102)	
Variable	Mean (SD) (*n*)	Mean (SD) (*n*)	Mean (SD) (*n*)	Mean (SD) (*n*)	Mean (SD) (*n*)	*p*-Value
Age (years)	66 (12.31) (407)	68.24 (11.23) (102)	66.99 (10.42) (102)	66.61 (12.09) (101)	62.16 (14.44) (102)	0.009
Female (%)	43.73% (178)	50.98% (52)	39.22% (40)	37.62% (38)	47.06% (48)	0.171
CKD etiology % (n)						0.426
Unknown cause	18.23% (406)	15.62% (15)	20.39% (21)	20% (21)	16.67(17)	
Diabetes mellitus	37.68% (153)	37.5% (36)	38.83% (40)	32.38% (34)	42.16% (43)	
APKD	1.72% (7)	3.12% (3)	0% (0)	2.86% (3)	0.98% (1)	
Cardiorenal syndrome	7.88% (32)	5.21% (5)	10.68% (11)	6.67% (7)	8.82% (9)	
Hypertension	23.4% (95)	30.21% (29)	20.39% (21)	20% (21)	23.53% (24)	
Glomerulonephritis	7.88% (32)	6.25% (6)	7.77% (8)	12.38% (13)	4.9% (5)	
Other etiology	3.2% (13)	2.08% (2)	1.94% (2)	5.71% (6)	2.94% (3)	
Protein intake (g/Kg/day)	0.74 (0.24) (407)	0.71 (0.28) (102)	0.7 (0.21) (102)	0.76 (0.23) (101)	0.79 (0.23) (102)	0.03
BMI (m^2^/kg)	28.24 (5.48) (407)	27.93 (6.08) (102)	28.9 (4.96) (102)	28.22 (5.19) (101)	27.9 (5.63) (102)	0.495
Systolic blood pressure (mmHg)	124.88 (17.39) (407)	125.23 (16.48) (102)	123.02 (17.72) (102)	125.74 (18.49) (101)	125.55 (16.93) (102)	0.674
Diastolic blood pressure (mmHg)	74.02 (9.09) (407)	73.24 (8.42) (102)	72.88 (11.01) (102)	74.19 (8.51) (101)	75.79 (7.94) (102)	0.079
Cardiovascular disease (%)	18.18% (74)	18.63% (19)	19.61% (20)	16.83% (17)	17.65% (18)	0.961
Diabetes mellitus (%)	37.59% (153)	37.25% (38)	34.31% (35)	38.61% (39)	40.2% (41)	0.847
Hypertension (%)	37.1% (151)	41.18% (42)	41.18% (42)	34.65% (35)	31.37% (32)	0.37
Hemoglobin (g/dL)	12.11 (1.54) (407)	11.99 (1.53) (102)	11.85 (1.43) (102)	12.5 (1.59) (101)	12.09 (1.57) (102)	0.021
Glucose (mg/dL)	123.27 (69.08) (407)	121.58 (37.96) (102)	114.62 (41.98) (102)	132.3 (115.6) (101)	124.69 (50.84) (102)	0.286
Albumin (g/dL)	3.83 (0.39) (407)	3.72 (0.41) (102)	3.83 (0.33) (102)	3.86 (0.37) (101)	3.89 (0.42) (102)	0.027
Azotemia (mg/dL)	92.44 (37.15) (407)	103.56 (35.82) (102)	95.27 (36.7) (102)	82.06 (34.15) (101)	88.75 (38.83) (102)	0.0002
Creatinine (mg/dL)	2.35 (0.91) (407)	2.69 (1.01) (102)	2.48 (0.97) (102)	2.07 (0.81) (101)	2.14 (0.71) (102)	<0.0001
Creatinine Clearance (mL/min)	32.07 (13.31) (407)	26.96 (12.91) (102)	30.6 (11.67) (102)	35.09 (12.05) (101)	35.64 (14.65) (102)	<0.0001
Phosphate (mg/dL)	3.88 (0.86) (407)	3.42 (0.68) (102)	3.82 (0.6) (102)	3.86 (0.73) (101)	4.42 (1.05) (102)	<0.0001
Calcium (mg/dL)	9.13 (0.6) (407)	9.05 (0.67) (102)	9.11 (0.64) (102)	9.16 (0.53) (101)	9.19 (0.53) (102)	0.392
PTH (pg/mL)	153.63 (124.71) (407)	166.31 (123.2) (102)	166.26 (128.84) (102)	122.37 (81.33) (101)	159.26 (151.17) (102)	0.003
Potassium (mEq/L)	4.9 (0.69) (407)	4.82 (0.7) (102)	4.96 (0.61) (102)	4.82 (0.67) (101)	4.98 (0.75) (102)	0.171
Sodium (mEq/L)	139.57 (3.16) (407)	139.54 (3.21) (102)	139.96 (3.01) (102)	139.33 (3.09) (101)	139.47 (3.33) (102)	0.49
Bicarbonate (mEq/L)	23.64 (3.06) (407)	23.59 (3.25) (102)	23.57 (3.22) (102)	23.91 (3.01) (101)	23.5 (2.79) (102)	0.766
Uric Acid (mg/dL)	5 (1.69) (407)	5.1 (1.76) (102)	5.04 (1.68) (102)	4.9 (1.54) (101)	4.97 (1.79) (102)	0.833
Cholesterol (mg/dL)	159.02 (35.57) (407)	158.75 (31.51) (102)	156.67 (32.8) (102)	154.37 (35.18) (101)	166.26 (41.36) (102)	0.156
LDL cholesterol (mg/dL)	91.65 (28.39) (407)	90.45 (22.22) (102)	89.04 (28.79) (102)	91.57 (29.93) (101)	95.52 (31.74) (102)	0.464
Triglicerides (mg/dL)	142.74 (60.57) (406)	146.36 (80.15) (102)	131.37 (48.65) (102)	138.11 (49.14) (101)	155.21 (56.92) (101)	0.013
C-reactive protein	5.93 (7.57) (406)	6.39 (8.34) (102)	5.78 (5.81) (102)	5.84 (8.24) (101)	5.69 (7.72) (101)	0.924
Homocystein	42.27 (48) (404)	43.15 (41.61) (101)	30.66 (24.26) (102)	38.31 (44.25) (100)	57.05 (68.17) (101)	0.001
24-h phosphaturia (mg/day)	518.67 (237.99) (407)	533.12 (249.77) (102)	501.85 (221.58) (102)	525.44 (226.65) (101)	514.35 (254.49) (102)	0.792
FePi (%)	44.19 (30.09) (407)	82.07 (34.03) (102)	45.68 (8.72) (102)	30.8 (6.7) (101)	18.08 (6.55) (102)	<0.0001
24 h sodiuria (mEq/day)	137.2 (53.95) (407)	132.4 (53.27) (102)	132.31 (47.5) (102)	144.25 (59.71) (101)	139.92 (54.52) (102)	0.326
24-h potassiuria (mEq/day)	43.49 (18.01) (407)	39.05 (15.21) (102)	43.94 (17.72) (102)	45.49 (17.49) (101)	45.51 (20.68) (102)	0.015
24-h proteinuria (mg/day)	484.14 (725.41) (407)	466.78 (784.05) (102)	488.08 (673.73) (102)	370.96 (485.15) (101)	609.64 (885.77) (102)	0.1

**Table 2 jcm-08-01026-t002:** predictors of Fractional Excretion of Phosphate (FeP) at linear regression analysis.

	Estimate	Std. Error	t Value	Pr(>|t|)	
(Intercept)	5.040	0.454	11.108	<2 × 10^−16^	***
Creatinine Clearance (mL/min)	−0.015	0.003	−5.574	0.000	***
age (years)	0.004	0.002	1.886	0.060	.
Protein Intake (g/Kg/day)	−0.170	0.128	−1.324	0.186	
Azotemia (mg/dL)	0.003	0.001	3.960	0.000	***
Diastolic Blood Pressure (mmHg)	−0.006	0.003	−2.001	0.046	*
Albumin (g/dL)	−0.047	0.074	−0.636	0.525	
Serum Phosphate (mg/dL)	−0.221	0.034	−6.479	0.000	***
Parathyroid hormone (pg/mL)	0.000	0.000	1.176	0.240	
Triglicerides (mg/dL)	0.000	0.000	0.685	0.494	
24-h potassiuria (mEq/day)	−0.001	0.002	−0.552	0.581	

Significance codes: 0 ‘***’ 0.001; ‘**’ 0.01; ‘*’ 0.05; ‘.’ 0.1; ‘ ’ 1.

**Table 3 jcm-08-01026-t003:** predictors of abnormal Fractional Excretion of Phosphate (>20%) at logistic regression analysis. Table legend: OR: odds ratio; LCI: lower bounder of 95% confidence interval; UCI: upper bounder of 95% confidence interval.

	OR	LCI	UCI	
Creatinine Clearance (mL/min)	0.995	0.992	0.998	**
age (years)	1.003	1	1.006	
Protein Intake (g/Kg/day)	0.919	0.781	1.082	
Azotemia (mg/dL)	1	0.999	1.001	
Diastolic Blood Pressure (mmHg)	0.995	0.992	0.999	*
Albumin (g/dL)	0.933	0.849	1.026	
Serum Phosphate (mg/dL)	0.939	0.899	0.981	**
Parathyroid hormone (pg/mL)	1	1	1	
Triglicerides (mg/dL)	1	0.999	1.001	
24-h potassiuria (mEq/day)	1	0.998	1.002	

Significance codes: 0 ‘***’ 0.001; ‘**’ 0.01; ‘*’ 0.05; ‘.’ 0.1; ‘ ’ 1.

**Table 4 jcm-08-01026-t004:** time to event according to quartiles of Fractional Excretion of Phosphate (FeP). (**A**) time to the composite endpoint of all-cause mortality or progression to ESRD; (**B**) time to the progression to ESRD; (**C**) time to all-cause mortality. Analysis for each endpoint are presented unadjusted, adjusted for the case-mix (age, CVD, Hypertension, Diabetes, Systolic and diastolic blood pressure, renal function) and fully adjusted (case-mix and protein intake, azotemia, albumin, serum phosphate, parathyroid hormone, triglicerides, 24-h urinary potassium).

**(A) Risk of ESRD or all-cause mortality according to quartile of FePi.**				
**Unadjusted**	**exp(coef)**	**lower. 95 u**	**upper. 95**	**Pr (>|z|)**
Low FePi	Ref	Ref	Ref	Ref
Mid-low FePi	2.06	1.29	3.30	0.003
Mid-High FePi	2.54	1.60	4.04	0.000
High FePi	2.75	1.74	4.34	0.000
**Case-Mix adjusetd**	**exp(coef)**	**lower. 95 u**	**upper. 95**	**Pr (>|z|)**
Low FePi	Ref	Ref	Ref	Ref
Mid-low FePi	1.80	1.12	2.90	0.015
Mid-High FePi	2.20	1.38	3.51	0.001
High FePi	2.42	1.50	3.91	0.000
**Fully adjusetd**	**exp(coef)**	**lower. 95 u**	**upper. 95**	**Pr (>|z|)**
Low FePi	Ref	Ref	Ref	Ref
Mid-low FePi	2.02	1.23	3.31	0.006
Mid-High FePi	2.29	1.42	3.71	0.001
High FePi	2.40	1.44	3.99	0.001
**(B) Risk of ESRD according to quartile of FePi**				
**Unadjusted**	**exp(coef)**	**lower. 95 u**	**upper. 95**	**Pr (>|z|)**
Low FePi	Ref	Ref	Ref	Ref
Mid-low FePi	2.32	0.70	7.70	0.170
Mid-High FePi	4.73	1.57	14.25	0.006
High FePi	8.46	2.96	24.20	0.000
**Case-Mix adjusetd**	**exp(coef)**	**lower. 95 u**	**upper. 95**	**Pr (>|z|)**
Low FePi	Ref	Ref	Ref	Ref
Mid-low FePi	2.31	0.69	7.71	0.174
Mid-High FePi	4.49	1.47	13.68	0.008
High FePi	6.47	2.16	19.41	0.001
**Fully adjusetd**	**exp(coef)**	**lower. 95 u**	**upper. 95**	**Pr (>|z|)**
Low FePi	Ref	Ref	Ref	Ref
Mid-low FePi	3.29	0.88	12.33	0.078
Mid-High FePi	7.34	2.14	25.16	0.002
High FePi	12.34	3.65	41.73	0.000
**(C) Risk of All-cause mortality according to quartile of FePi**				
**Unadjusted**	**exp(coef)**	**lower. 95 u**	**upper. 95**	**Pr (>|z|)**
Low FePi	Ref	Ref	Ref	Ref
Mid-low FePi	2.01	1.21	3.36	0.008
Mid-High FePi	2.16	1.29	3.61	0.004
High FePi	1.75	1.02	2.99	0.041
**Case-Mix adjusetd**	**exp(coef)**	**lower. 95 u**	**upper.95**	**Pr (>|z|)**
Low FePi	Ref	Ref	Ref	Ref
Mid-low FePi	1.43	0.85	2.41	0.175
Mid-High FePi	1.79	1.06	3.01	0.030
High FePi	1.45	0.83	2.56	0.193
**Fully adjusetd**	**exp(coef)**	**lower. 95 u**	**upper. 95**	**Pr (>|z|)**
Low FePi	Ref	Ref	Ref	Ref
Mid-low FePi	1.44	0.84	2.49	0.187
Mid-High FePi	1.68	0.98	2.88	0.058
High FePi	1.33	0.72	2.46	0.359

Model 1: unadjusted; Model 2: adjusted for case-mix (age, CVD, Hypertension, Diabetes, Systolic and diastolic blood pressure, renal function); Model 3: adjusted for model 2 and protein intake, azotemia, albumin, serum phosphate, PTH, triglicerides, 24-h urinary potassium; NB variable forced into the model are selected apriori + variable associated with FePi at univariable analysis (*p* < 0.10).

**Table 5 jcm-08-01026-t005:** Most parsimonious model (logistic regression analysis) to predict CKD progression.

	Estimate	Std. Error	z Value	Pr (>|z|)	
(Intercept)	0.58	1.99	0.29	0.771	
High FeP % (>75th percentile of study cohort distribution)	1.42	0.37	3.79	0.000	***
Age (years)	−0.04	0.01	−3.18	0.001	**
History of diabetes (yes vs. no)	0.66	0.34	1.94	0.052	.
Diastolic blood pressure (mmHg)	−0.03	0.02	−1.86	0.063	.
Azotemia (mg/dL)	0.01	0.00	2.46	0.014	*
Serum Phsophate (mg/dL)	0.48	0.19	2.48	0.013	*
24-urine potassium	0.02	0.01	2.13	0.033	*
Protein Intake (g/Kg/day)	−1.17	0.36	−3.25	0.001	**

Significance codes: 0 ‘***’ 0.001; ‘**’ 0.01; ‘*’ 0.05; ‘.’ 0.1; ‘ ’ 1.

**Table 6 jcm-08-01026-t006:** Model discrimination with and without FeP to predict CKD progression.

Model	Estimate
Most parsimonious model without high-Fep	
C-statistic (95% CI)	0.770 (95% CI: 0.695–0.846)
Most parsimonious model with high-Fep	
C-statistic (95% CI)	0.802 (95% CI: 0.734–0.869)
Comparison of models	
Chi-square (*p* value)	10.9 (0.0009)
IDI (95% CI) (*p* value)	0.027 (−0.004−0.0544) (0.053)
Continuous NRI (95% CI) (*p* value)	0.46(0.18−0.74) (0.001)

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
