# Peer review of "Fractional Excretion of Phosphate (FeP) Is Associated with End-Stage Renal Disease Patients with CKD 3b and 5"

_jcm, 2019, doi:10.3390/jcm8071026_

Round 1
Reviewer 1 Report
Antonio et al., have done a cross-sectional, retrospective study in 407 CKD 3b- 4 patients to evaluate the phosphate balance and outcomes.
The authors have done an excellent job in planning and executing the study. The manuscript prepared well.
Here are my comments,
1.Title:
- Could be more specific.
Can consider’ Fractional excretion of Phosphate is associated with End-stage renal disease patients with CKD 3b and 4’.
2. Abstract:
In methods, authors wrote to study 407 patients of CKD 3b to 4, and in the results, they mention about CKD 3 to 5. Please, correct.
3. Experimental Section:
- Data collection:
Authors need to be more precise on the timeline of the study. How long were the patients followed?
Inclusion and exclusion criteria?
Etiology of the CKD? As different etiologies for the CKD’s can have different progression to ESRD.
Dietary practices of the included patient population?
The laboratory which was collected, was it done at the beginning of the study and if they were repeated? If only one sample is obtained, then how does it will impact the outcomes when compared to multiple samples?
One sample of FeP obtained at the time of referral, without prior knowledge of the dietary restrictions could have influenced by the dietary practices of the patients at the time. The dietary practices could have changed following their evaluation and education provided by the Nephrologists resulting in a change in the FeP.
More samples or a sample, particularly after the nephrologist and dietary education could have made a difference? Need authors comment.
Why were FGF-23 levels were not studied?
- The authors wrote,’ History of cardiovascular disease (CVD) was defined as a history of myocardial infarction, stroke, coronary artery by-pass surgery, angioplasty, and peripheral arterial disease.’
How about the hx of HFpEF or HFrEF? Were any of the patients had Echocardiogram’s done or available?
If not, please include it in the study limitations.
-Table 1- Please clearly mention the parameters of the different quartiles of the FeP.
What is the 5th vertical column in table 1, which is written as Quartile (102)?
How was the daily protein intake of patients in table 1, calculated?
4.Discussion:
Authors wrote, ‘no association between 24-h urine phosphate and the renal outcome has been documented’. Prakobsuk et al., in their 2017 study demonstrated the same in renal transplant recipients (PMID: 27981393)
Last but not least, the manuscript will benefit from a thorough English language editing, given multiple errors.
Author Response
Dear Editor,
I am submitting the revised version of the above referenced manuscript to your attention for potential publication in the Journal of Clinical Medicine.
On behalf of my coworkers, thank you very much for the constructive advices provided to us. I am re-submitting the above referenced manuscript after having made several revisions in accordance with your reviewers’ and editor’s suggestions.
Enclosed, are (1) edited as well as a (2) clean version of the revised manuscript. In the edited version, the amendments are highlighted in blue and strikethrough font if deletion and in red font if addition. Furthermore, here below, please find a point by point responses to the reviewers’ comments.
We hope you will now find the manuscript ready for acceptance although I remain available for further modifications as you might deem necessary.
Warm regards
Biagio Di Iorio
Point-by-point answer to reviewers:
Reviewer #1
Bellasi Antonio et al., have done a cross-sectional, retrospective study in 407 CKD 3b- 4 patients to evaluate the phosphate balance and outcomes. The authors have done an excellent job in planning and executing the study. The manuscript prepared well.
Answer: we thank the reviewer for the praise.
Here are my comments:
Title: Could be more specific. Can consider’ Fractional excretion of Phosphate is associated with End-stage renal disease patients with CKD 3b and 4’.
Answer: we thank the reviewer for the suggestion. The title has been amended accordingly.
Abstract: in methods, authors wrote to study 407 patients of CKD 3b to 4, and in the results, they mention about CKD 3 to 5. Please, correct.
Answer: we apologize for the lack of clarity. Stage 5 patients were also included in current analyses. We rectified title and text accordingly
Experimental Section:
Data collection: Authors need to be more precise on the timeline of the study. How long were the patients followed?
Answer: as specified in the text: “For this retrospective study, an historical cohort of 407 records of CKD stage 3b-5 patients attending between January 2010 and October 2015 at the Nephrology Unit of Solofra (AV), Italy were utilized”. Furthermore, in the results section, we specified that the “mean follow-up was 45 months (min: 3, max 60)”.
Inclusion and exclusion criteria?
Answer: No specific inclusion and exclusion criteria were enforced. We added a sentence in the text “All consecutive adult patients referred to the Unit were included in this retrospective analysis” to better clarify this point.
Etiology of the CKD? As different etiologies for the CKD’s can have different progression to ESRD.
Answer: we have now added this piece of information in table 1. FeP is not associated with CKD etiology. Because the strength of the association at univariable analysis does not meet the a priori criterium of p<.10, CKD etiology was not forced in the survival models.
Dietary practices of the included patient population?
Answer: this piece of information is not available. We have acknowledged it as a study limitation.
The laboratory which was collected, was it done at the beginning of the study and if they were repeated? If only one sample is obtained, then how does it will impact the outcomes when compared to multiple samples?
Answer: we thank the reviewer for this interesting question. For the purpose of this analyses we only collected one laboratory data at the beginning of the study. We have acknowledged it as a study limitation.
One sample of FeP obtained at the time of referral, without prior knowledge of the dietary restrictions could have influenced by the dietary practices of the patients at the time. The dietary practices could have changed following their evaluation and education provided by the Nephrologists resulting in a change in the FeP.
More samples or a sample, particularly after the nephrologist and dietary education could have made a difference? Need authors comment.
Answer: this is another interesting question. However, this piece of information is not available nor we have data on FeP changes after the first visit at the Nephrology Unit. We have acknowledged it as a study limitation.
Why were FGF-23 levels were not studied?
Answer: this is a retrospective study and we don’t have data on FGF-23 levels. We have acknowledged it as a study limitation.
The authors wrote,’ History of cardiovascular disease (CVD) was defined as a history of myocardial infarction, stroke, coronary artery by-pass surgery, angioplasty, and peripheral arterial disease.’
How about the hx of HFpEF or HFrEF? Were any of the patients had Echocardiogram’s done or available? If not, please include it in the study limitations.
Answer: this is a retrospective study and we don’t have these data. We have acknowledged it as a study limitation.
Table 1- Please clearly mention the parameters of the different quartiles of the FeP.
What is the 5th vertical column in table 1, which is written as Quartile (102)?
Answer: we apologize for the lack of clarity. We edited table 1 adding the cutoff values of FeP and specifying that the 5th vertical column represent the high quartile of the FeP distribution. Finally we added a table legend to increase the readability of the table.
How was the daily protein intake of patients in table 1, calculated?
Answer: as specified in the text “Protein intake was calculated according to the following formula: Protein intake = 6.25*(0.46*UUN + 0.031*ideal body weight) + PPU (where UUN is urinary urea and PPU is proteinuria).”
4.Discussion:
Authors wrote, ‘no association between 24-h urine phosphate and the renal outcome has been documented’. Prakobsuk et al., in their 2017 study demonstrated the same in renal transplant recipients (PMID: 27981393)
Answer: this piece of information has now been added. We thank the reviewer for the remark.
Last but not least, the manuscript will benefit from a thorough English language editing, given multiple errors.
Answer: the text has been revised. We thank the reviewer for the remark.
Reviewer #2
Bellasi et al. investigated the association of fractional excretion of phosphate and the mortality or ESRD in CKD patients. The authors nicely showed the possible usefulness of fractional excretion of phosphate as a marker for CKD progression.
Answer: we thank the reviewer for the praise
I had no specific suggestions for the authors except for the following points:
The progression of chronic kidney disease depends in part on the cause of the CKD. Please include the etiology of the CKD in patient characteristics.
Answer: we have now added this piece of information in table 1. FeP is not associated with CKD etiology. Because the strength of the association at univariable analysis does not meet the a priori criterium of p<.10, CKD etiology was not forced in the survival models.
The serum phosphate levels increases in advanced CKD, however phosphate levels in the low quartile group (this group showed lowest creatinine clearance compared to the other groups) are relatively low. Did the authors investigate the use of phosphate lowering medications?
Answer: this is a retrospective study and we don’t have these data. We have acknowledged it as a study limitation.
Patients with CKD stage 3b-4 were included, however the baseline renal function below CCr<15 is documented.
Answer: we apologize for the lack of clarity. Stage 5 patients were also included in current analyses. We rectified title and text accordingly
Reviewer 2 Report
Comments and Suggestions for Authors
Bellasi et al. investigated the association of fractional excretion of phosphate and the mortality or ESRD in CKD patients. The authors nicely showed the possible usefulness of fractional excretion of phosphate as a marker for CKD progression. I had no specific suggestions for the authors except for the following points:
The progression of chronic kidney disease depends in part on the cause of the CKD. Please include the etiology of the CKD in patient characteristics.
The serum phosphate levels increases in advanced CKD, however phosphate levels in the low quartile group (this group showed lowest creatinine clearance compared to the other groups) are relatively low. Did the authors investigate the use of phosphate lowering medications?
Patients with CKD stage 3b-4 were included, however the baseline renal function below CCr<15 is documented.
Author Response

(The authors gave the same response as above.)

Round 2
Reviewer 1 Report
I have reviewed the manuscript before, and authors have responded to comments and suggestions to my satisfaction.
I have the following comments.
1.Title:
Please consider modifying it to ‘Fractional excretion of Phosphate (FeP) is associated with End-stage renal disease in CKD stage 3b to 5 patients.’
2.Abstract:
Still common English language errors.
It's fractional and not fraction in line 4 of the background
3.Introduction:
Please include a brief paragraph on the FeP with its normal values, clinical importance, and latest evidence.
What is the reason to include elaborate information on the role and influence of the RAS blockade and hyperphosphatemia? In the introduction.
4.Materials and Methods:
Please include the details on the patient consent and IRB approval number if available.
5.Outcome assessment:
How are the outcome 2 and three are different from outcome 1?
6.Results:
Authors have used CKD stage 3 to 5 multiple times in the manuscript when the study population in CKD stage 3b and 5. Please amend.
The authors made a decent effort to current the grammatical mistakes, but there are still quite a few, which needs attention. Please have it reviewed with a native English language expert before resubmission.
Author Response
Dear Editor,
I am submitting the revised version of the above referenced manuscript to your attention for potential publication in the Journal of Clinical Medicine.
On behalf of my coworkers, thank you very much for the constructive advices provided to us. I am re-submitting the above referenced manuscript after having made several revisions in accordance with your reviewers’ and editor’s suggestions.
Enclosed, are (1) edited as well as a (2) clean version of the revised manuscript. In the edited version, the amendments are highlighted in blue and strikethrough font if deletion and in red font if addition. Furthermore, here below, please find a point by point responses to the reviewers’ comments.
We hope you will now find the manuscript ready for acceptance although I remain available for further modifications as you might deem necessary.
Warm regards
Biagio Di Iorio
Point-by-point answer to reviewers:
Reviewer #1
Have reviewed the manuscript before, and authors have responded to comments and suggestions to my satisfaction.
Answer: we thank the reviewer for the praise.
I have the following comments.
1.Title: Please consider modifying it to ‘Fractional excretion of Phosphate (FeP) is associated with End-stage renal disease in CKD stage 3b to 5 patients.’
Answer: title has been edited accordingly
2.Abstract:
Still common English language errors.
It's fractional and not fraction in line 4 of the background
Answer: as per the reviewer suggestion we edited line 4 and the text accordingly.
3.Introduction:
Please include a brief paragraph on the FeP with its normal values, clinical importance, and latest evidence.
Answer: a paragraph on FeP has been added.
What is the reason to include elaborate information on the role and influence of the RAS blockade and hyperphosphatemia? In the introduction.
Answer: we have removed this part form the introduction
4.Materials and Methods: Please include the details on the patient consent and IRB approval number if available.
Answer: in light of the retrospective nature of the study, patients’ data were retrieved form medical records, deidentified and imported in the study database. The study protocol was notified to the local IRB. According to the local laws, no formal approval of such studies is required.
5.Outcome assessment: How are the outcome 2 and three are different from outcome 1?
Answer: the question is not clear. However, we tested the association of FeP and 3 different outcome: (1) occurrence of the composite endpoint of ESRD and all-cause mortality; (2) occurrence of ESRD; (3) occurrence of all-cause mortality. All-cause mortality and ESRD are regarded as the most clinically relevant outcomes for CKD patients. However, in light of the association between fast renal function deterioration and risk of death, it is important to test the effect of FeP on the combination of the two outcomes before testing the effect of FeP on each of the two outcomes separately. We edited the text to clarify this point.
6.Results:
Authors have used CKD stage 3 to 5 multiple times in the manuscript when the study population in CKD stage 3b and 5. Please amend.
Answer: CKD stage has been amended throughout the text as suggested
The authors made a decent effort to current the grammatical mistakes, but there are still quite a few, which needs attention. Please have it reviewed with a native English language expert before resubmission.
Answer: the text has been revised. We thank the reviewer for the remark.